# Expression of SIGLEC15 correlates with tumor immune infiltration, molecular subtypes, and breast cancer progression

**Huan Lai[1], Yiyang Liu[1], Yan Gong[1], Chuanyu Zong[2], Wei Zeng[3]\*, Honglei Chen[1,2]\***

1 Department of Pathology, Zhongnan Hospital of Wuhan University, Wuhan, P. R. China, 2 Department of Pathology, School of Basic Medical Sciences, Wuhan University, Wuhan, P. R. China, 3 Department of Radiation and Medical Oncology, Zhongnan Hospital of Wuhan University, Wuhan, Hubei, P. R. China

\* zengweixh@126.com (WZ); hl-chen@whu.edu.cn (HC)

**Data Availability Statement:** The bioinformatical data underlying the results presented in the study are available from TCGA, etc. In our manuscript,

## Abstract

Breast cancer (BRCA) is among the most prevalent cancers and is responsible for numerous patient fatalities. Immunotherapy has emerged as a promising approach to cancer treatment. Recent studies have identified Siglec-15 as a novel immune target that plays a crucial role in tumor immune evasion, suggesting its potential significance in BRCA. We utilized databases such as TCGA to investigate the relevance of SIGLEC15 in BRCA. The expression of the Siglec-15 protein in 74 breast cancer patients was detected using immunohistochemistry, and its association with clinicopathological features and overall survival was evaluated. The co-expression of Siglec-15, CD68, CK, and CD8 in BRCA tissues was identified through multiplex immunofluorescence staining. Our study revealed that SIGLEC15 expression in BRCA was significantly elevated compared to adjacent normal tissues. Kaplan-Meier analysis identified SIGLEC15 as a prognostic protective factor. According to the receiver operating characteristic curve analysis, SIGLEC15 could predict the luminal subtype of BRCA. Enrichment analysis demonstrated that SIGLEC15 involves various biological pathways, including immunity, metabolism, tumors, and infectious diseases. Correlation analysis revealed an association between SIGLEC15 expression and immune infiltration in BRCA. We also confirmed that the Siglec-15 protein is expressed in cancer cells, tumor-infiltrating T cells, and macrophages in BRCA tissues, significantly higher levels than in normal breast tissues. Consequently, SIGLEC15 correlates with tumor immune infiltration, molecular subtypes, and BRCA progression and prognosis. However, further research is required to elucidate the role of SIGLEC15 in breast cancer.

## 1. Introduction

Breast cancer (BRCA) is among the most common types of cancer. In 2020, it was anticipated that there would be 2.3 million new cases (11.7%) of cancer worldwide, making female BRCA the most frequently diagnosed disease, surpassing lung cancer. The same year saw 685,000 deaths from breast cancer, ranking it the fifth leading cause of cancer-related deaths globally

we showed the website,such as: https://xenabrowser.net/datapages/.

**Funding:** The author(s) received no specific funding for this work.

[1]. Recent statistical studies have shown an increase in both the incidence and mortality rates of BRCA in China, with 117,174 fatalities and 416,371 new cases reported in 2020 [2]. BRCA is a collection of diseases with distinct pathological characteristics and clinical outcomes. Current clinical practice categorizes BRCAs into five subtypes based on histological (hormone receptor expression) and molecular (Ki-67 proliferation marker index) features: HER2 enriched, triple-negative, luminal A-like, luminal B-like HER2-, and luminal B-like HER2+ [3]. The mainstream treatments for breast cancer remain surgery, chemotherapy, and radiation therapy [4]. Thanks to advances in these treatments, the mortality rate among BRCA patients decreased by 40% from 1979 to 2017 [5]. However, approximately 10% of BRCA patients still succumb to chemotherapy-resistant metastases and inoperable diseases [6]. Recently, immunotherapy has emerged as a promising new direction for cancer treatment by modulating the tumor micro-environment (TME).

TME comprises immune cells, mesenchymal cells, tumor cells, and secreted cytokines and chemokines. These components interact to regulate the physiological function of tumor cells [7]. Immune checkpoints are crucial targets for preventing excessive immunity, but tumor cells can exploit these checkpoints to evade immune surveillance. Blocking the tumor's immune checkpoints can alter the TME and prevent tumor immune escape. The success of immune checkpoint inhibitors (ICIs) and chimeric antigen receptor (CAR) T-cell therapies against various cancers led to the recognition of cancer immunotherapy as a scientific break-through in 2013 [8]. Programmed death-ligand 1 (PD-L1) is one of the most extensively uti-lized immune checkpoints. It is primarily expressed on tumor cells and a small number of tumor-infiltrating lymphocytes (TILs). In BRCA, approximately 5% of patients with triple-negative breast cancer (TNBC) express PD-L1. Although PD-L1 ICIs show some promise when combined with standard chemotherapy for advanced and metastatic TNBC [9], it is essential to acknowledge that some patients exhibit either primary or acquired resistance to immunotherapy medications [10]. Furthermore, PD-L1 expression is less common in non-TNBC than in TNBC [9, 11, 12]. Therefore, identifying new immune checkpoints for the immunotherapy of BRCA patients is of utmost importance.

The previous study showed that sialoglycans interact with sialic acid-binding immunoglob-ulins, such as sialic acid-binding immunoglobulin-type lectins (Siglecs), to modulate tumor immunity [13]. Among Siglecs, Siglec-15 is considered an important and unique molecule due to its special structure and prominent role in various biological processes [14]. Siglec-15 is a type I transmembrane protein containing only one V-set immunoglobulin (Ig) domain and one C2 set immunoglobulin, highly similar to PD-L1 [15]. Its role in cancer immune regula-tion was also confirmed [15]. Siglec-15 is expressed in many human cancer cells, and tumor-infiltrating immune cells, and is significantly associated with poor outcomes in human solid tumors [16]. Targeting Siglec-15 may be a successful immunotherapy strategy for patients who are not responding to anti-PD-L1, as Siglec-15 expression is mutually exclusive with PD-L1 expression and independent of the PD-L1/PD-1 pathway in lung adenocarcinoma [15]. Siglec-15 expression in macrophages prevents antigen-specific T lymphocytes from proliferating, which promotes tumor growth [15]. The significant inhibitory effect of Siglec-15 on antigen-specific T-cell responses and the role of initiating immune evasion in TME has been demon-strated *in vivo* and *in vitro* [17]. Mechanistic studies have shown that the interaction between Siglec-15 expressed in macrophages and lung cancer cells expressing sialyl-Tn antigen (sTn) enhances the secretion of TGF-β, which leads to immunosuppression through the DAP12/Syk pathway [18, 19]. In phase I clinical study for advanced non-small cell lung cancer (NSCLC), patients receiving Siglec-15 inhibitors (NC318) showed potential effectiveness (NCT03665285). Moreover, Siglec-15 is clinically up-regulated in renal clear cell carcinoma and pancreatic ductal carcinoma [20, 21].

In this study, we employed the TCGA, Genotype-Tissue Expression (GTEX), UCSC XENA, and Kaplan Meier (KM) plotter to examine the difference in SIGLEC15 expression, the clinical importance of SIGLEC15, and the correlation between SIGLEC15 and clinicopathological characteristics. Siglec-15 protein expression in BRCA and its predictive relevance were confirmed by immunohistochemistry (IHC) and multiplex immunohistochemical staining (mIHC). Also, we looked into SIGLEC15's gene enrichment and investigated its prognostic value in immune infiltration. In addition, our study evaluated the expression of Siglec-15 protein in 74 cases of human BRCA tissues and analyzed its relationship with clinicopathological features and overall survival (OS) in patients.

## 2. Materials and methods

### 2.1 Data processing and differential expression, survival, and correlation analysis

We utilized the TCGA dataset and the GTEx databases to collect raw counts of RNA-sequencing data and accompanying clinical information for tumor tissues and adjacent tissues from breast cancer. All analyses were conducted using R software, version 3.6.3. The R packages "ggplot2", "survminer," and "survival" were employed to plot expression analysis and KM survival curves. Univariate Cox proportional hazards regression was used to calculate p-values and the hazard ratio (HR) with a 95% confidence interval (CI) in KM curves. Log-rank tests were utilized to determine significance. The R package "ggstatsplot" was used for the two-gene correlation analysis. Pearson's correlation or Spearman's correlation analysis was applied to assess the correlation between quantitative variables. The R package "pROC" was used to analyze the diagnostic significance of SIGLEC15 mRNA. The ssGSEA algorithm provided in R Package "GSVA" was used to calculate immune infiltration using the markers of 24 immune cells [22].

### 2.2 UCSC XENA

We downloaded RNAseq data in tpm format for TCGA and GTEx that were uniformly processed by the Toil process from UCSC XENA (https://xenabrowser.net/datapages/) [23].

### 2.3 TNMplot.com analysis platform

TNMplot.com Analysis Platform (TNMplot, https://tnmplot.com/analysis/) includes 56,938 unique samples from GEO, GTex, TCGA, and TARGET databases [24]. We used the data from TNMplot.com Analysis Platform to explore the expression of the SIGLEC15.

### 2.4 Tumor Immune Single-cell Hub 2

Tumor Immune Single-cell Hub 2 (TISCH2, http://tisch.comp-genomics.org/home/) is a scRNA-seq database focusing on tumor microenvironment [25]. We used TISCH2 to study the expression of SIGLEC15 at the single-cell level.

### 2.5 Molecular Taxonomy of Breast Cancer International Consortium

Molecular Taxonomy of Breast Cancer International Consortium B (METABRIC) is a large study involving 2,000 breast cancer patients [26]. Based on its data, we explored the impact of SIGLEC15 on OS.

## 2.6 Kaplan Meier plotter

The KM plotter (http://kmplot.com/analysis/) is capable of assessing the correlation between the expression of all genes (mRNA, miRNA, protein) and survival in 30k+ samples from 21 tumor types including breast, ovarian, lung, and gastric cancer [27]. Sources for the databases include GEO, EGA, and TCGA. We used KM plotter to explore the prognostic value of SIGLEC15 in different immune cell subgroups in BRCA.

## 2.7 LinkedOmics database

We collected the co-expression genes of SIGLEC15 in BRCA from the LinkedOmics database (http://www.linkedomics.org/login.php) to perform gene set enrichment analysis [28]. The signaling pathway of SIGLEC15 in BRCA was analyzed for Kyoto Encyclopedia of Genes and Genomes (KEGG) enrichment using Gene Set Enrichment Analysis (GSEA)software and the ClusterProfiler package. Gene ontology (GO) enrichment and KEGG pathway analyses of co-expression genes, depicted by the R package "ggplot2," were performed using the R program "ClusterProfiler."

## 2.8 Tissue microarray construction and patients' follow-up

Two separate breast cancer tissue microarrays (TMAs) were procured and utilized in this study (sourced from Guilin Fanpu Biotechnology Co., China). The TMA slides comprised 75 breast cancer tissues and corresponding non-cancerous tissues collected between June 2006 and November 2014. All 150 cores, each with a diameter of 1.5 mm, were arranged in paraffin blocks in the TMAs. The histological diagnosis and levels of differentiation [29] were assessed using the 2019 WHO classification of breast cancers. Due to insufficient tumor cells in one tissue sample, it was excluded, leaving the tumor tissues of the remaining 74 patients for evaluation of Siglec-15 expression. Additional clinicopathological characteristics recorded are detailed in S1 Table in S1 Data, including age, pathology type, survival status, pathological grading, intrinsic molecular subtype, tumor size (T), lymph node metastasis (N), and distant metastasis (M).

The follow-up period commenced post-surgery and concluded in March 2015. OS was defined as the time from initial diagnosis to either death or the end of follow-up. Patients who died due to unforeseen circumstances or other diseases were excluded from the survival cohorts. The average overall survival period for patients with breast cancer was 45 months, ranging from 4 to 105 months.

All procedures involving human participants adhered to the ethical standards of Guilin Fanpu Biotechnology Co(No:2018023). All research complied with the principles of the 1964 Declaration of Helsinki and its later amendments or comparable ethical standards. Because of this type of retrospective study, informed consent is not required. All data/samples were fully anonymized before we accessed them.

## 2.9 Immunohistochemistry analysis

IHC analysis was conducted to detect the expression of Siglec-15 protein in BRCA tissues. Initially, TMA sections were deparaffinized and rehydrated. Following this, antigen retrieval for Siglec-15 was performed in EDTA (1 mM, pH 8.0) buffer for 15 minutes using a microwave. The sections were then incubated with a rabbit anti-human Siglec-15 antibody (1:100 dilution; Thermo Fisher Scientific, USA) at 4°C overnight. Subsequently, the TMAs were incubated successively with a biotinylated secondary antibody and horseradish peroxidase (HRP)-conjugated streptavidin (UltraSensitive™ SP Detection System, Maixin, Fuzhou, China) at 37°C for 30 minutes. This was followed by chromogen development with 3,3'-diaminobenzidine (DAB; Dako, Agilent, USA) and nuclear counterstaining with hematoxylin.

## 2.10 Evaluation of immunohistochemistry

Light microscopy was used to detect immunostaining reactivity (Olympus BX-53 with CCD DP74). Two pathologists, who were independent and blinded to the clinicopathological aspects of the study, scored the results. An agreement was reached by comparing the two pathologists' scores.

The expression of Siglec-15 protein was mainly evaluated in cancer cells and immune cells with the semi-quantitation method by the area of positive (AP) and the intensity of staining (IS). AP was graded as follows: 0 (0–5%), 1 (6–25%), 2 (26–50%), 3 (51–75%), and 4 (>75%). IS was graded as 0 (negative), 1 (weak), 2 (moderate), and 3 (strong). Siglec-15 expression was calculated based on the equation: Intensity distribution (ID) = AP × IS [30]. The X-Tile determined that 4.5 was the best cut-off point for Siglec-15 expression [31], ID≥4.5 meant high expression of Siglec-15 protein, on the contrary, ID<4.5 meant low expression of Siglec-15 protein.

## 2.11 Multiplex immunohistochemical staining

mIHC staining was performed on 4-μm sections obtained from 10 cases of formalin-fixed paraffin-embedding (FFPE) breast cancer blocks using the Opal 5-Color Kit (Akoya Biosciences, MA, USA). These FFPE breast cancer blocks were collected from Department of Pathology, Zhongnan Hosipital of Wuhan University. Ethical approval for our study was obtained from the Medical Ethics Committee, Zhongnan Hospital of Wuhan University (No:2017057). The stained slides were scanned using a Vectra multispectral microscope (Akoya Biosciences, MA, USA). The immunofluorescence markers used were CD68 (pre-diluted, 1:1 dilution), CK (prediluted, then 1:2 dilution), CD8 (prediluted, then 1:2 dilution) from Agilent Technologies Inc., and Siglec-15 (1:200 dilution; Thermo Fisher Scientific, USA).

Each primary antibody was visualized using tyramide signal amplification linked to a specific fluorochrome from the Opal 5-Color Kit. Based on the Meidi microwave (Meidi, China), a stripping procedure was performed for each consecutive antibody staining. The mIHC-stained slides and uniplex IF-stained slides were scanned with a Vectra 3.0 microscope system with InForm 2.6 software (Akoya Biosciences, MA, USA) under fluorescent illumination. The wavelengths corresponding to the four markers were CD68 (wavelength 520 nm), CK (wavelength 620 nm), CD8 (wavelength 690 nm), and Siglec-15 (wavelength 570 nm).

## 2.12 Statistical analysis

All statistical analyses were conducted using SPSS 22.0 software (Chicago, IL, USA) and R (V 3.6.3). The significance of SIGLEC15 expression between tumor and normal tissues was determined using the Wilcoxon rank sum test. The relationship between Siglec-15 protein expression and associated clinicopathological variables was examined using the $\chi 2$ test or Fisher's exact test. OS was estimated using the Kaplan Meier method and the log-rank test. Univariate and multivariate Cox proportional hazard regression models were used to investigate the independent prognostic impacts on survival. Significance levels were set at P<0.05 (*), P<0.01 (**), and P<0.001 (***), respectively.

## 3. Results

### 3.1 SIGLEC15 mRNA expression in BRCA

We investigated the mRNA expression levels of SIGLEC15 in BRCA using the TCGA, GTEX, TNMplot, and TICSH2 databases. The adjacent tissues of TCGA and the normal tissues of GTEX were integrated and labeled as normal tissues, then the normal and tumor tissues were compared (Fig 1A). The result revealed that SIGLEC15 mRNA expression levels were elevated

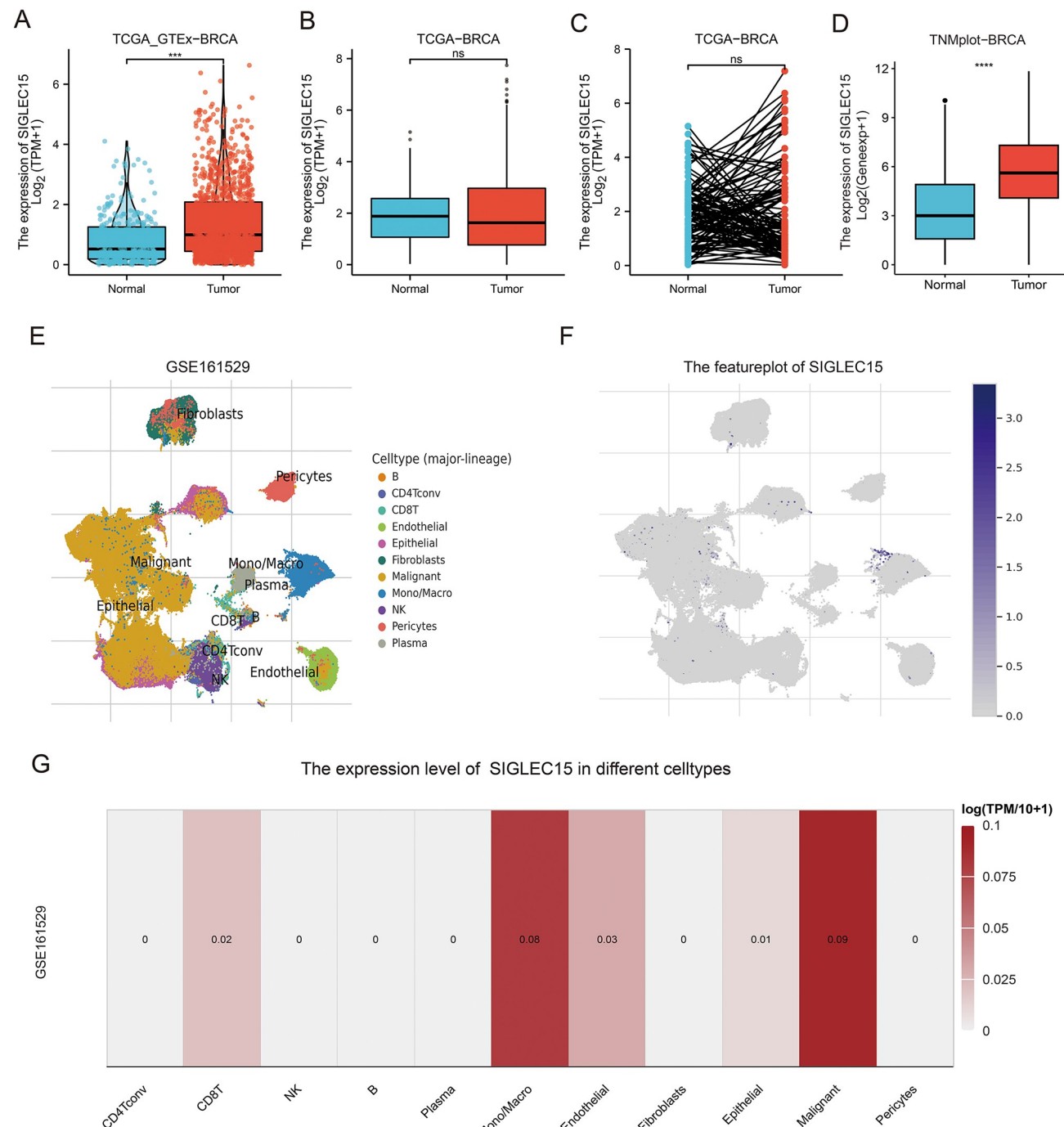

**Fig 1. SIGLEC15 expression in the transcriptome and single-cell transcriptome in BRCA.** (A) SIGLEC15 mRNA expression in breast cancer tissues and normal tissues from TCGA and GTEx databases; (B) SIGLEC15 mRNA expression in breast cancer tissues and adjacent normal tissues from TCGA database; (C) SIGLEC15 mRNA expression in breast cancer patients and matched adjacent normal samples in TCGA; (D) SIGLEC15 mRNA expression in tumor and normal tissues after removing adjacent non-tumor tissue in TNMplot data; (E) Cell subtypes of breast cancer at the level of single-cell; (F) The featured plot of SIGLEC15 mRNA expression at the level of single-cell; (G) SIGLEC15 expression in different cell types.

in tumor tissues compared to normal tissues (Fig 1A). No discernible difference between tumor and normal tissues was observed in the TCGA data (Fig 1B and 1C). According to TNMplot's data, significantly high expression of SIGLEC15 in tumors could be observed after

removing adjacent samples in the normal group (Fig 1D). Additionally, by analyzing single-cell databases from TICSH2, we found that SIGLEC15 is primarily expressed in malignant cells and monocytes/macrophages (Fig 1E–1G).

## 3.2 SIGLEC15 mRNA expression was associated with different clinical parameters in BRCA

We examined the association between SIGLEC15 mRNA expression and various clinicopathological parameters, such as ER status, PR status, HER2 status, PAM50 molecular subtype, TNM stage, pathological stage, and histological type. Our analysis revealed that a high expression of SIGLEC15 was linked to ER and PR receptors (Fig 2A and 2B). However, it showed no association with the expression of HER2 (Fig 2C). We also found that SIGLEC15 mRNA was most expressed in breast cancer with the luminal subtype and least in the basal-like subtype (Fig 2D and 2E). Furthermore, a significant difference in the expression of SIGLEC15 was observed only between the T1 and T2 stages (Fig 2F–2H). Regarding the pathological stage, SIGLEC15 expression was higher in stage I compared to other stages (Fig 2I and 2J). Notably, SIGLEC15 mRNA expression significantly correlated with the histological type (Fig 2K).

## 3.3 Diagnostic significance of SIGLEC15 mRNA for molecular subtypes

We assessed the diagnostic value of SIGLEC15 for molecular subtypes by generating ROC curves. The predictive ability of the variable SIGLEC15 for ER-Positive and Negative outcomes has a certain level of accuracy (AUC = 0.727, Fig 2L). When dividing BRCA into PR-positive and PR-negative groups, the AUC is 0.702 (Fig 2M). In predicting LumA & LumB and Her2 & Basal outcomes, the variable SIGLEC15 had a certain level of predictive accuracy with an AUC of 0.750 (Fig 2N). In predicting outcomes for infiltrating ductal carcinoma and infiltrating lobular carcinoma, the variable SIGLEC15 demonstrated lower accuracy with an AUC of 0.657 (Fig 2O).

## 3.4 Overexpression of SIGLEC15 mRNA was a protective factor in BRCA

We utilized data from TCGA, METABRIC and KM plotter to study the prognostic value of SIGLEC15 mRNA expression in BRCA. In TCGA, high expression of SIGLEC15 was associated with better OS (S1A Fig in S1 Data). These results were also validated in METABRIC (S1B Fig in S1 Data) and the integrated gene chip data from KM plotter (S1C Fig in S1 Data). In GSE7390, which studied lymph node-negative breast cancer, we found that high expression of SIGLEC15 is a protective factor (S1D Fig in S1 Data). The data from GSE20685 and GSE45255 also confirmed that overexpression of SIGLEC15 is advantageous for OS (S1E, S1F Fig in S1 Data). The dataset GSE48390 study also confirmed that high expression of SIGLEC15 is a protective factor of patients with Han Chinese breast cancer (S1G Fig in S1 Data).

## 3.5 Prognosis value of SIGLEC15 mRNA in different clinical subgroups

We divided breast cancer patients into various subgroups based on different clinical parameters to further explore the clinical significance of SIGLEC15. We then examined the impact of SIGLEC15 on the survival curve of each subgroup. In TCGA data, our findings revealed that higher levels of SIGLEC15 were associated with extended survival in several categories, including those with T1-T2 stage, age ≤60, no radiation therapy, M0 stage, stage I-II, and stage II (S2A-S2F Fig in S1 Data).

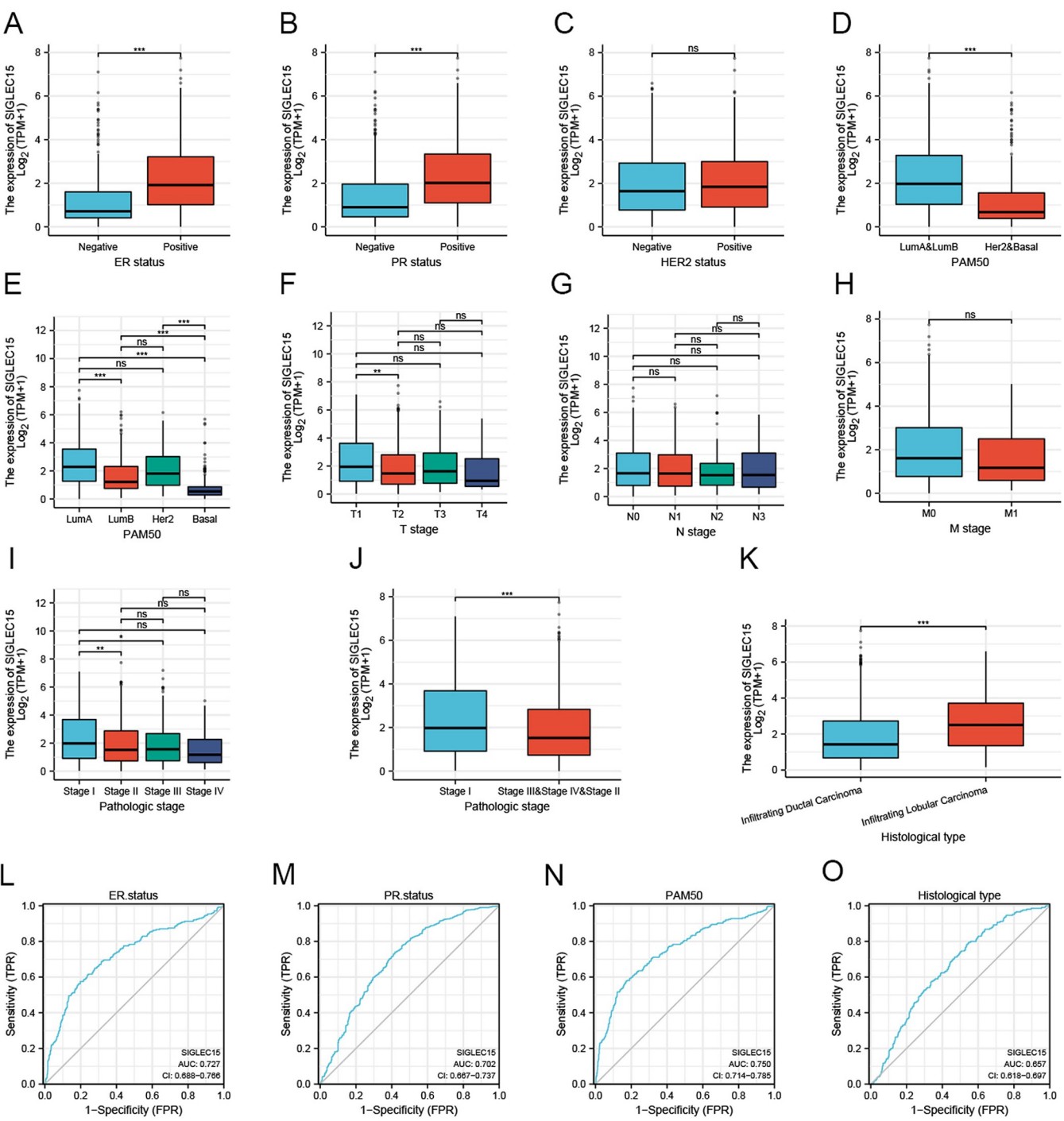

**Fig 2. Correlation between SIGLEC15 mRNA and clinicopathological parameters and ROC curves.** (A) ER status; (B) PR status; (C) HER2 status; (D, E) PAM50; (F-H) TNM stage; (I, J) Pathological stage; (K) Histological type; (L) SIGLEC15 predicts ER receptor expression levels in breast cancer patients; (M) SIGLEC15 predicts PR levels in breast cancer patients; (N) SIGLEC15 predicts the PAM50 molecular subtypes in breast patients; (O) SIGLEC15 predicts pathological types in breast cancer patients.

### 3.6 Relationship between SIGLEC15 mRNA and tumor immunity

The invasion of tumors by immune cells can influence the prognosis of cancer. Using the TCGA database, we examined if there was a correlation between SIGLEC15 mRNA and the

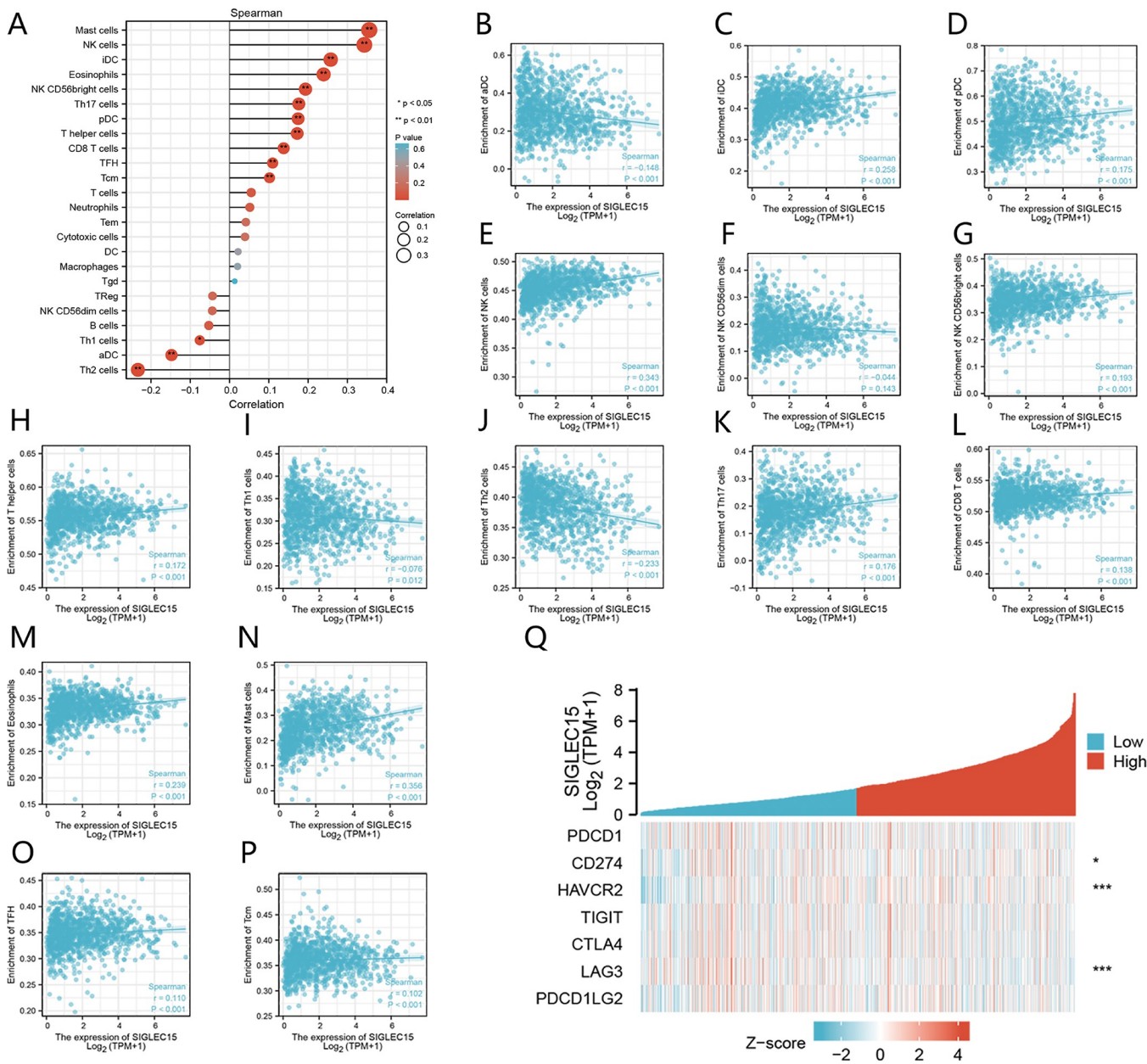

**Fig 3. Correlation analysis of SIGLEC15 mRNA expression and infiltration levels of immune cells in BRCA.** (A-P) Correlation between SIGLEC15mRNA expression levels and immune cell infiltration levels; (Q) Correlation analysis of SIGLEC15 expression with immune checkpoint-associated genes in BRCA.

degree of immune infiltration in BRCA. SIGLEC15 showed a positive association with mast cells, NK cells, immature DC, eosinophils, NK CD56 bright cells, Th17 cells, plasmacytoid DC, T helper cells, CD8+ T cells, T follicular helper, and T central Memory. There was a negative association of SIGLEC15 with Th2 cells, activated DC, and Th1 cells (Fig 3A–3P). This study indicated that SIGLEC15 acts as a modulatory molecule within the immune microenvironment.

Given SIGLEC15's emerging role as a potential immune checkpoint (IC), exploring its relationship with other known ICs was important. Our analysis revealed a correlation between SIGLEC15 and established ICs such as CD274/PD-L1, HAVCR2, and LAG3 (Fig 3Q).

### 3.7 The prognostic relevance of SIGLEC15 mRNA based on immune cells in BRCA

Earlier research indicated a positive correlation of SIGLEC15 with most immune cells. To determine if SIGLEC15 might influence patient prognosis through its effect on immune cell infiltration, we employed the KM plotter to gauge the impact of SIGLEC15 on patient outcomes across various immune subgroups. Our findings suggested that elevated SIGLEC15 mRNA expression was associated with improved prognosis in subsets with diminished CD8 + T cells, type 1 T helper cells, type 2 T helper cells, eosinophils, macrophages, and in the enriched NK cell subgroup (S3A-S3L Fig in S1 Data). This study provides evidence that SIGLEC15 might enhance survival rates in breast cancer patients by modulating specific immune cell infiltration.

### 3.8 Co-expressed genes of SIGLEC15 and functional enrichment analysis in breast cancer from the LinkedOmics database

We utilized the LinkedOmics database to identify a significant correlation with the SIGLEC15 gene. Spearman correlation analysis revealed the correlated genes of SIGLEC15 (S4A Fig in S1 Data). The heatmap displays 50 genes positively or negatively associated with SIGLEC15 (S4B, S4C Fig in S1 Data). GO and KEGG analyses revealed that these genes were involved in various biological processes and pathways such as organelle fission, nuclear division, chromosome segregation, control of cell cycle phase transition, and mitotic nuclear division (S4D Fig in S1 Data).

SIGLEC15 was also found to be associated with the chromosomal regions, spindles, chromosomes, centromeric regions, condensed chromosomes, microtubules, ATPase activity, tubulin binding, microtubule binding, ATPase activity, and catalytic activity on DNA (S4E, S4F Fig in S1 Data). KEGG molecular pathways included cell cycle, oocyte meiosis, human T-cell leukemia virus 1 infection, and progesterone-mediated oocyte maturation (S4G Fig in S1 Data).

### 3.9 Gene set enrichment analysis of SIGLEC15-related signaling pathways

We conducted a GSEA pathway enrichment analysis to explore the biological function of SIGLEC-15. The results indicated that SIGLEC-15 primarily involves several biological processes and pathways, including E2F-targets, G2M checkpoint, pancreatic beta cells, myc-target-V1, KRAS-SIGNALING-DN, and spermatogenesis. This was determined by evaluating the differentially expressed genes between the high-expression and low-expression SIGLEC-15 groups (S4H-S4M Fig in S1 Data).

### 3.10 Expression of Siglec-15 protein in the human breast cancer

We further validated the expression of Siglec-15 protein in breast cancer at the histological level. Distinct regional patterns of Siglec-15 positivity were observed in both human breast cancer and adjacent non-cancerous breast tissues. The Siglec-15 protein was predominantly localized in the cytoplasm, with occasional presence in both the cytoplasm and nucleus of cancer cells (Fig 4, S5B Fig in S1 Data). We noted that Siglec-15 expression was either absent or faint in the normal breast gland epithelium (Fig 4A and 4F), prominently stained in the immune cells, especially in stromal macrophages (Fig 4C, S5A, S5C Fig in S1 Data). In breast cancers with varied pathological classifications, Siglec-15 protein staining was identified in the invasive and *in situ* ductal carcinoma (Fig 4B–4E). The mIHC results highlighted co-expression of Siglec-15 with CK, CD68, and CD8 (Fig 5). These findings underscored that the Siglec-

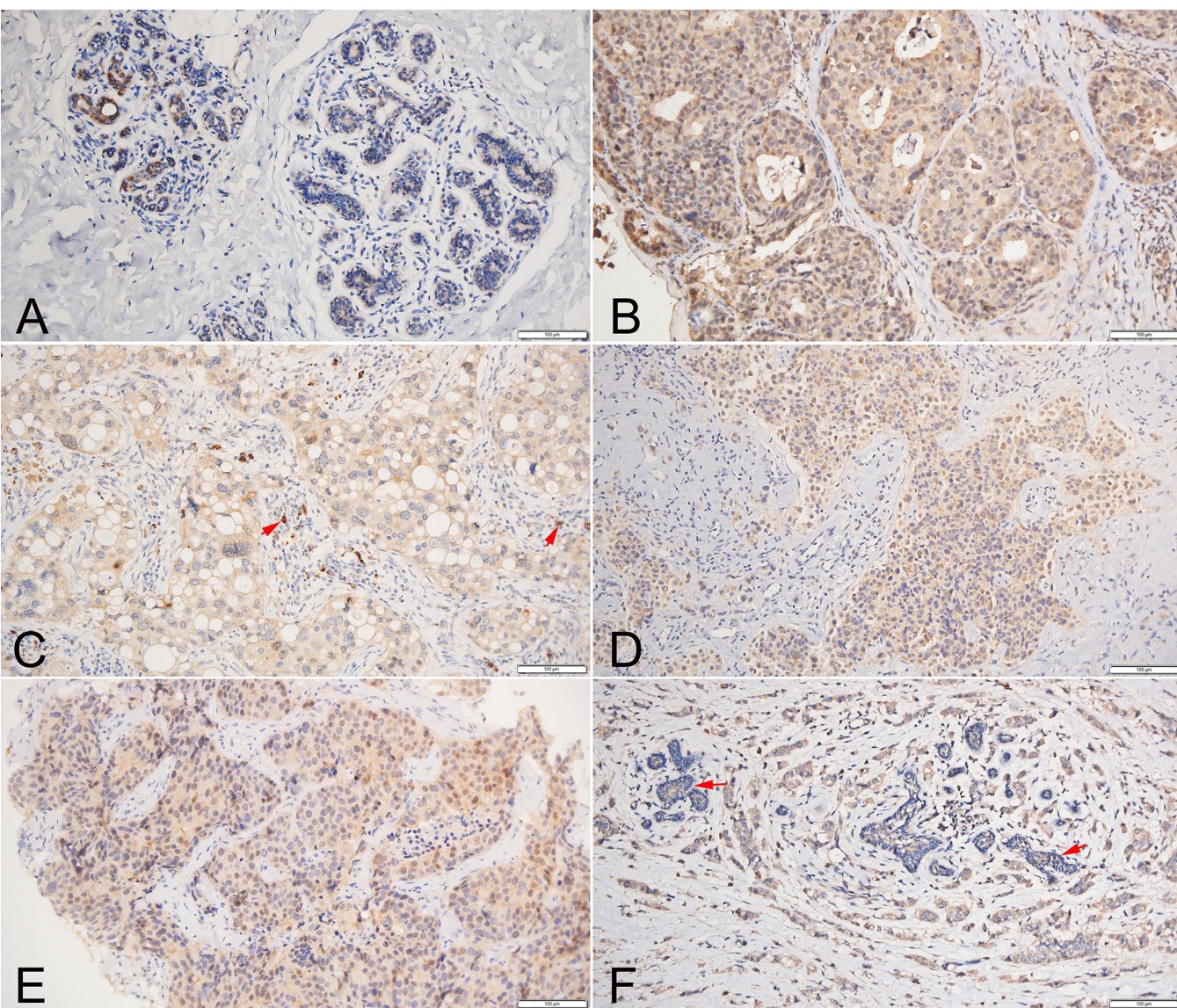

**Fig 4. Representative photomicrographs of Siglec-15 protein expression in human BRCA tissues.** (A) Displayed faint or absent expression of Siglec-15 protein in the normal breast gland epithelium. (B) Highlighted positive staining of Siglec-15 protein in breast ductal in situ carcinoma. (C) Illustrated positive staining of Siglec-15 protein in the cytoplasm of invasive ductal carcinoma and in the immune cells of the stroma (indicated by the arrow). (D) Showed perinuclear positivity for Siglec-15 protein in invasive ductal carcinoma. (E) Revealed positive staining of Siglec-15 protein in both the cytoplasm and nucleus of invasive ductal carcinoma. (F) Indicated the absence of Siglec-15 expression in the normal breast gland epithelium (indicated by the arrow), while positive signals were evident in the invasive ductal carcinoma.

15 protein is present in cancer cells, tumor-infiltrating T cells, and macrophages within breast cancer tissues. Moreover, high expression of Siglec-15 protein was observed in 54 out of 74 cancer tissues, a rate significantly greater than the 6 out of 74 observed in corresponding non-cancerous breast tissues ($P<0.001$).

### 3.11 Clinicopathological significance of Siglec-15 expression in the breast cancer

We assessed the relationship between Siglec-15 expression and the clinicopathological characteristics of breast cancer using the chi-square test. As delineated in Table 1, among breast

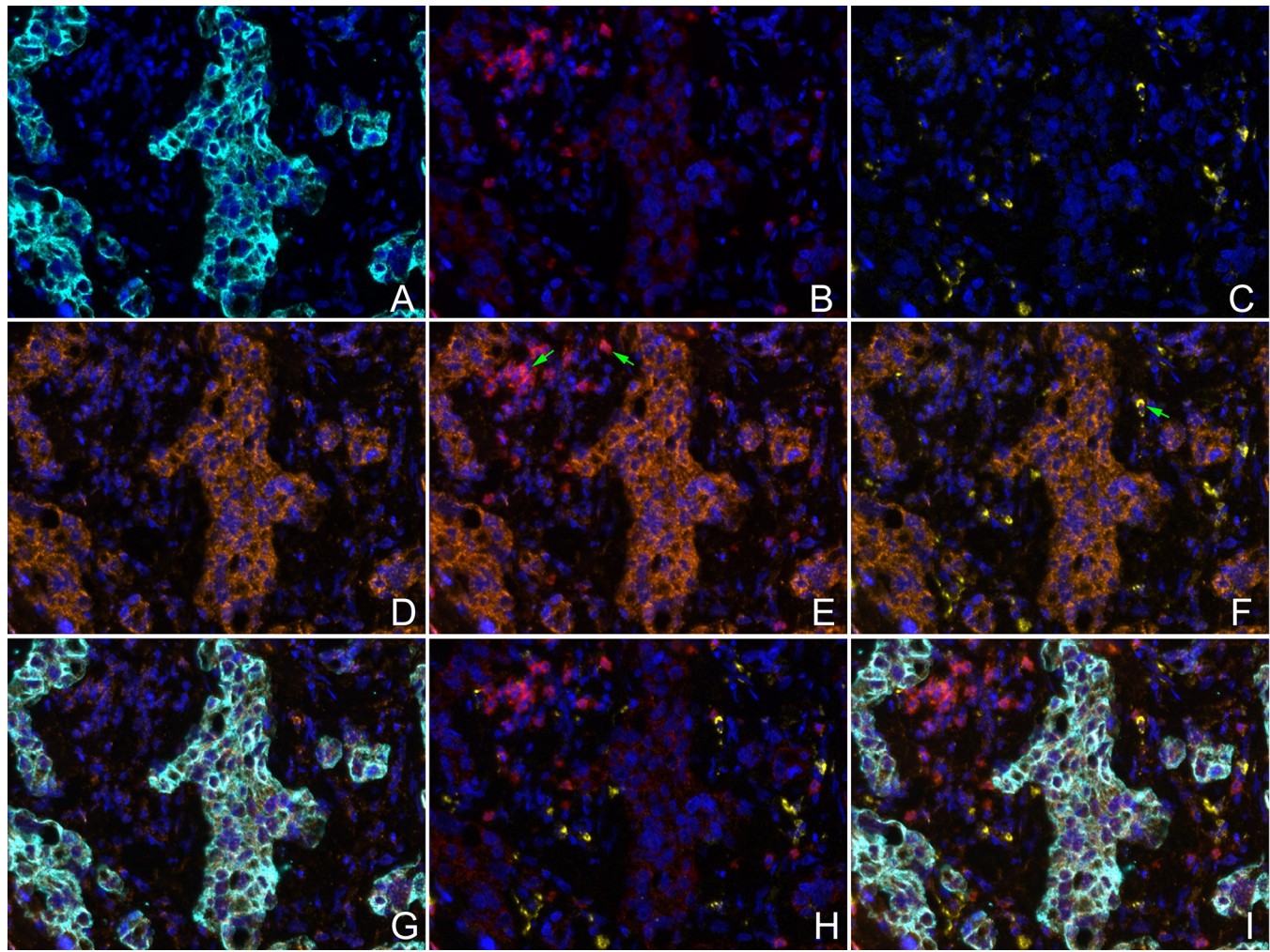

**Fig 5. CK, Siglec-15, CD8, and CD68 proteins detected by mIHC in the human BRCA tissues.** (A) CK; (B) CD8; (C) CD68; (D) Siglec-15; (E) co-expression of Siglec-15 and CD8 (green arrow); (F) co-expression of Siglec-15 and CD68 (green arrow); (G) co-expression of Siglec-15 and CK; (H) no co-expression of CD8 and CD68; (I) merged images of all markers. (Original magnification A-I ×400).

cancer patients, Siglec-15 expression in tumor cells was strongly associated with pathological subtypes (P = 0.004) and pathological grades (P<0.001). Notably, the prevalence of Siglec-15 expression was greater in pathological grades I and II compared to grade III. While the difference was not statistically significant, an observable trend suggested that Siglec-15 protein expression in luminal A and luminal B subtypes was higher than in HER2-enriched and triple-negative subtypes. For instance, high expression of Siglec-15 was observed in 75% (12/16) of luminal A cases and 57.1% (8/14) of triple-negative breast cancer cases.

### 3.12 Prognostic value of Siglec-15 protein in breast cancer patients

We subsequently evaluated the prognostic value of Siglec-15 through KM curves, with comparisons made using the log-rank test. Within the breast cancer cohort, a high expression of Siglec-15 in tumor cells was indicative of a more favorable overall survival (OS); however, the difference did not reach statistical significance (P = 0.052) (S5D Fig in S1 Data). Continuing our analysis, we evaluated the prognostic significance of Siglec-15 alongside other

**Table 1. Associations of Siglec-15 protein with clinicopathological parameters of breast cancer tissues.**

| Parameters | Cases | Siglec-15 in cancer cells | | |
|---|---|---|---|---|
| | | Low (%) | High (%) | P value |
| Age | | | | 0.702 |
| > = 45 | 38 | 11(29.0%) | 27(71.0%) | |
| <45 | 36 | 9(25.0%) | 27(75.0%) | |
| Pathological subtype | | | | **0.004** |
| Invasive ductal carcinoma | 65 | 14(21.5%) | 51(78.5%) | |
| Invasive lobular carcinoma | 9 | 6(66.7%) | 3(33.3%) | |
| Pathological grade | | | | <**0.001** |
| I, II | 63 | 13(20.6%) | 50(79.4%) | |
| III | 11 | 7(63.6%) | 4(36.4%) | |
| Tumor size (T) | | | | 0.103 |
| T1, T2 | 48 | 10(20.8%) | 38(79.2%) | |
| T3, T4 | 26 | 10(38.5%) | 16(61.5%) | |
| Lymph node metastasis (N) | | | | 0.439 |
| N0 | 46 | 11(33.9%) | 35(76.1%) | |
| N1, N2 | 28 | 9(32.1%) | 19(67.9%) | |
| TNM stage | | | | 0.307 |
| I, II | 61 | 15(24.6%) | 46(75.4%) | |
| III | 13 | 5(38.5%) | 8(61.5%) | |
| Intrinsic molecular subtype | | | | 0.064 |
| Luminal A, luminal B | 46 | 9(19.6%) | 37(80.4%) | |
| HER2-enriched, triple-negative | 28 | 11(39.3%) | 17(60.7%) | |

clinicopathological factors using Cox proportional hazards models. In the univariate analysis, factors such as tumor size (P = 0.031, HR: 0.300, 95%CI: 0.101–0.897), lymph node metastasis (P = 0.004, HR: 0.180, 95%CI: 0.056–0.577), and TNM stage (P = 0.010, HR: 0.248, 95%CI: 0.085–0.720) demonstrated significant correlations with the survival rate of breast cancer patients. While not reaching statistical significance (P = 0.063, HR: 0.364, 95%CI: 0.126–1.058), a high expression of Siglec-15 protein in tumor cells suggested a favorable prognosis. In the multivariate analysis, only lymph node metastasis (P = 0.007, HR: 0.175, 95%CI: 0.049–0.623) exhibited a significant association with the survival rate of breast cancer patients, marking it as an independent prognostic factor (Table 2).

## 4. Discussion

In our study, we analyzed the expression patterns of SIGLEC15 mRNA in BRCA using the TCGA, GTEx, TISCH2, and TNMplot databases. Although the analysis using only TCGA data did not show a difference in the expression of SIGLEC15 mRNA between tumor and normal groups, the inclusion of GTEx data allowed us to observe high expression of SIGLEC15 mRNA in tumor tissues. Additionally, using TNMplot to study the expression differences between healthy and tumor samples after excluding adjacent non-tumor tissues also revealed high expression of SIGLEC15 mRNA. Moreover, single-cell data confirmed the high expression of SIGLEC15 mRNA in malignant cells. Therefore, we believe that SIGLEC15 mRNA is highly expressed in BRCA. Concurrently, associations emerged between SIGLEC15 mRNA and clinical attributes, including pathological type, clinical stage, molecular type, as well as ER and PR receptor expression levels. The ROC curve analysis also indicated the potential utility of SIGLEC15 mRNA in identifying the luminal subtype, which exhibited a superior OS relative to the non-luminal subtype.

**Table 2. COX proportional hazard models on overall survival of breast cancer patients.**

| Factors | Univariate analysis | | Multivariate analysis | |
|---|---|---|---|---|
| | P value | HR (95%CI) | P value | HR (95%CI) |
| Expression of Siglec-15 | **0.052** | 0.364(0.126–1.058) | 0.335 | 0.508 (0.073–1.489) |
| High vs. Low | | | | |
| Age | 0.494 | 1.447(0.502–4.173) | 0.343 | 1.705 (0.566–5.136) |
| ≥45 vs.<45 | | | | |
| Pathological subtype | 0.989 | 0.990(0.221–4.432) | 0.833 | 1.214 (0.201–7.349) |
| IDC vs. ILC | | | | |
| Pathological grade | 0.790 | 0.815(0.182–3.651) | 0.468 | 1.907 (0.333–10.920) |
| I+II vs. III | | | | |
| Tumor size (T) | **0.031** | 0.300(0.101–0.897) | 0.063 | 0.332 (0.104–1.063) |
| T1+T2 vs. T3+T4 | | | | |
| Lymph node metastasis (N) | **0.004** | 0.180(0.056–0.577) | **0.007** | 0.175 (0.049–0.623) |
| N0 vs. N1+N2 | | | | |
| TNM stage | **0.010** | 0.248(0.085–0.720) | 0.211 | 0.209 (0.018–2.437) |
| I+II vs. III | | | | |
| Intrinsic molecular subtype | 0.270 | 0.551(0.191–1.588) | 0.381 | 0.582 (0.183–1.914) |
| Luminal A+Luminal B vs. HER2-enriched+Triple-negative | | | | |

To corroborate these findings, we employed IHC and found elevated Siglec-15 protein expression in 73.0% (54/74) of breast cancer tissues-a stark contrast to the 8.1% (6/74) expression observed in adjacent breast tissues. Further assessment using mIHC staining confirmed the expression of Siglec-15 protein in both cancer cells and immune cells. Moreover, a marked association was evident between high expression of Siglec-15 protein and pathological subtypes, and grades, as determined by IHC. While analyzing patients' follow-up data, we noted a trend toward improved outcomes in the high-expression cohort. However, this was not statistically significant. Interestingly, the TCGA and METABRIC databases did highlight a statistically significant prognostic role for SIGLEC15 mRNA. We hypothesize that this discrepancy arose due to the limited sample sizes in our patient follow-up dataset, potentially leading to subtle variations in survival curves.

Previous studies have indicated that SIGLEC15 mRNA plays a role in pan-cancer pathways related to immunity, metabolism, cancer, and infectious diseases, as demonstrated through GSEA [32]. Our study reaffirms these findings, showcasing SIGLEC15's involvement in similar pathways based on our GO, KEGG, and GESA analyses. Typically, Siglec-15 expression is restricted to cells originating from the bone marrow. In various cancers, the production of soluble mucoproteins could attenuate the immune response by interacting with Siglec receptors in natural killer cells, dendritic cells, and monocytes [33].

However, our findings present a different narrative. In our study, SIGLEC15 mRNA emerged as a prognostic marker for breast cancer patients. Within BRCA, SIGLEC15 mRNA displayed a positive correlation with the infiltration levels of multiple immune cells, suggesting its role in bolstering the immune microenvironment. Subsequent analysis of immune infiltration sub-groups also revealed that high expression of SIGLEC15 mRNA is a harbinger of improved OS. Another study demonstrated that Siglec-15 protein expression has no significant correlation with favorable prognosis in OS of 245 BRCA cases [34]. On the contrary, high expression of Siglec-15 in tumor tissues and peritumoral macrophages endow with an immunosuppressive microenvironment, so predicts poor survival in glioma patients [35]. Contrary to general assumptions, our data suggest that SIGLEC15 might act as a beneficial factor that

positively modulates immune responses in the context of breast cancer. In nasopharyngeal carcinoma patients, Siglec-15 expression has been correlated with improved distant failure-free survival (D-FFS) and independent predictors of D-FFS [36]. From the above, SIGLEC15 has an opposite predictor for the prognosis of different cancer subtypes, further research needs to confirm it.

This study sheds light on the relatively unexplored role of SIGLEC15 in BRCA. To bolster our findings, we employed IHC techniques on a breast cancer TMA and utilized mIHC to examine the co-expression patterns of Siglec-15 protein across various immune cells present in breast cancer. We found that Siglec-15 protein was mainly localized in the cytoplasm of cancer cells and immune cells, especially macrophages of the stroma. This is consistent with the subcellular localization of Siglec-15 in the breast cancer [34, 37]. However, Siglec-15 protein was occasionally presented in both the cytoplasm and nucleus of cancer cells, need more cases to confirm their clinicopathological significance. Although, high expression of Siglec-15 was easily detected in the invasive ductal BRCA and low-grade BRCA. Contrariwise, overexpression of Siglec-15 protein in tumor cells is associated with advanced TNM stage of colorectal cancer and predicts fewer tumor-infiltrating lymphocytes [38].

In summary, our research reveals SIGLEC15 as a protective agent in breast cancer progression, with the potential to predict the luminal subtype and low grade of breast cancer, especially the invasive ductal subtype. Crucially, we verified the expression of Siglec-15 in T cells and macrophages within the breast cancer tumor microenvironment, underscoring its immunomodulatory function. However, the role of SIGLEC15 in breast cancer progression needs to be investigated and confirmed.

## Supporting information

**S1 Data.**
(DOCX)

## Acknowledgments

We thank Medjaden Inc. for its assistance in the preparation of this manuscript.

## Author Contributions

**Data curation:** Wei Zeng.

**Formal analysis:** Yiyang Liu.

**Investigation:** Wei Zeng, Honglei Chen.

**Methodology:** Huan Lai, Yiyang Liu, Yan Gong, Chuanyu Zong.

**Supervision:** Honglei Chen.

**Validation:** Honglei Chen.

**Writing – original draft:** Huan Lai, Chuanyu Zong.

**Writing – review & editing:** Wei Zeng, Honglei Chen.

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
