## [Decision Letter · Decision Letter 0]

23 May 2024

PONE-D-24-03046Expression of SIGLEC15 correlates with tumor immune infiltration, molecular subtypes, and breast cancer progressionPLOS ONE

Dear Dr. Chen,

Thank you for submitting your manuscript to PLOS ONE. After careful consideration, we feel that it has merit but does not fully meet PLOS ONE’s publication criteria as it currently stands. Therefore, we invite you to submit a revised version of the manuscript that addresses the points raised during the review process.

**Please address all reviewers' comments below. Please also remove the S5 image from the Human Protein Atlas, or alternatively, include staining or higher resolution images to support that your staining is in either the nucleoplasm or Golgi. Please also use the Roman numerals for referring to tumour grade. Further to reviewer 2's comment - gene/mRNA should be italicised and capitalised, protein only capitalised. Lowercase is reserved for mouse proteins.**

We look forward to receiving your revised manuscript.

Kind regards,

Amy McCart Reed

Academic Editor

PLOS ONE

Journal Requirements:

2. Please provide additional details about the tissue blocks obtained. Specifically, please ensure that you have discussed whether all data/samples were fully anonymized before you accessed them.

3. Thank you for uploading your study's underlying data set. Unfortunately, the repository you have noted in your Data Availability statement does not qualify as an acceptable data repository according to PLOS's standards.

4. Please amend the manuscript submission data (via Edit Submission) to include author Dr. Chuanyu Zong.

Reviewers' comments:

Reviewer's Responses to Questions

**Comments to the Author**

1. Is the manuscript technically sound, and do the data support the conclusions?

Reviewer #1: Partly

Reviewer #2: Yes

2. Has the statistical analysis been performed appropriately and rigorously? 

Reviewer #1: Yes

Reviewer #2: Yes

3. Have the authors made all data underlying the findings in their manuscript fully available?

Reviewer #1: No

Reviewer #2: Yes

4. Is the manuscript presented in an intelligible fashion and written in standard English?

Reviewer #1: No

Reviewer #2: Yes

5. Review Comments to the Author

**Reviewer #1:** In their manuscript, Lai et al. investigate the significance of SIGLEC15 in breast cancer progression. By integrating public datasets and patient samples, they explore its expression across various subtypes, elucidate its influence on immune cell dynamics within the tumor microenvironment, and discuss its clinical implications. However, the current version of the Review falls short on many aspects which are listed below.

1.While the manuscript presents a number of interesting findings, there are several instances where these results are not fully explained or contextualized and inappropriate words have been used such as what´s more, which could limit the reader's understanding and the study's overall impact. Below are some examples that illustrate this point, though they are not exhaustive:

- The text and figure legends of the manuscript currently do not adequately explain the findings, leading to potential misinterpretation of the data. For instance, the presentation of Figure 3A is currently mislabeled as a heatmap. Additionally, the resolution or format of Figure 3A needs improvement, as the gene labels are currently illegible. Figure 3 A-C appear to represent differential expression analysis and not functional enrichment analysis (Figure D-J are functional enrichment analysis).

-The organization of subsections within the manuscript currently disrupts the logical flow of the presented data. For instance, the data from Figure 1d is discussed separately from its subsequent analysis related to the clinical parameters of BCa in TCGA, which appears later in Section 3.6. To improve the coherence and progression of the narrative, it is recommended to reorder the sections and join some of the subsections. Moreover, the allocation of data between supplementary and main sections does not appear to be based on the significance or importance of the data.

-How is the immune fraction in TCGA datasets analyzed? If tools like CIBERSORT or other deconvolution methods were employed but not explained. Also in section 3.5, the authors do not properly explain the method they used to do the analysis in the body text or the figure legend.

-The distinctions between tumor tissues and adjacent normal tissues from TCGA, and normal tissues from GTEx are not clearly made in the text and figure legends.

-The transition between public datasets to patient data is not clear.

2. A significant overlap between some of the analyses and figures presented in this manuscript and those published in the study by Baihui Li et al. (PMID: 32939323) is noticed. An example is Figure S1 and Figure 1.

3. In Figures S1 B and C, the analysis shows no significant difference in SIGLEC15 expression between normal and tumor tissues in BCa using TCGA as a reference dataset. Despite the lack of significant differences, the manuscript proceeds to analyze variations between different BCa subtypes in TCGA. The authors should use other commonly used public datasets such as METABRIC, lu et al. PMID: 18297396, Wang et al. PMID: 15721472 to confirm the overexpression of SIGLEC15.

4. In Figure 1E-J, the authors have utilized datasets with specific patient characteristics, such as GSE7390 and GSE65194, which provide data on gene expression changes due to therapies. Additionally, GSE20685 offers insights into environmental or lifestyle factors, like smoking, that may impact BCa at the molecular level. GSE4525 explores the genetic differences between hereditary and sporadic breast cancer cases. However, the manuscript does not discuss these distinctive characteristics when presenting the data, nor does it address their impact on survival outcomes between the low and high SIGLEC15 expression groups. These specific factors could significantly influence the survival rates observed. For instance, the effects of therapy might explain better survival, even when the overexpressed gene contributes to tumor progression. The authors should use other commonly used public datasets such as METABRIC, lu et al. PMID: 18297396, Wang et al. PMID: 15721472 to confirm the OS.

5. Given that the majority of analyses in this manuscript utilize publicly available datasets primarily at the mRNA level, such as TCGA, it is important to consider extending these analyses to include protein expression data. Protein differential analysis could be particularly insightful, as protein expression does not always correlate directly with mRNA levels due to post-transcriptional and post-translational modifications.

6. In Figure 4, the IHC sections illustrating SIGLEC-15 expression are currently not presented with sufficient clarity. Each section that demonstrates the expression of SIGLEC-15 should be paired with a corresponding negative control section to clearly establish baseline staining. Additionally, the transitions between ductal in situ carcinoma and ductal invasive carcinoma is not distinctly marked in the current presentation. Moreover, macrophages in tumor stroma have distinct markers that help in their identification. Without the use of these specific markers, distinguishing stromal macrophages from other cell types in the stroma based solely on morphology in IHC is inaccurate. The authors address this in Figure 5, but it can not be concluded from figure 4. Additionally, in Figure 5, the authors have not provided explanations regarding the cellular identities associated with the CK, CD68, or CD8 markers.

7. The study utilizes tissue microarray samples from 75 breast cancer patients to evaluate Siglec-15 protein expression, with histological examples provided. Although Siglec-15 protein levels are presented in Table 1, this association is not explicitly mentioned in the text with refferng to the corresponding data. Additionally, while patient samples are integral to the findings, their significance has not been adequately emphasized in both the introduction and discussion sections. Moreover, mRNA expression levels of Siglec-15 in the same patients have not been showcased. To improve data presentation, I recommend depicting both protein and mRNA expression levels in separate bar or dot graphs.

Based on my comments, I advise against publishing the manuscript in its current form. Significant revisions are necessary, both in terms of content and structure, to meet the standards of comprehensiveness and clarity required for publication.

**Reviewer #2:** The manuscript had a logical flow and was easy to follow with some minor issues that can be fixed. Overall, the result figures and the supplementary figures were well put and addressed. A few suggestions:

1. As a general comment for the document, the authors should standardize where they are using “SIGLEC15” and “Siglec-15” as it is not consistent in the document. From my understanding SIGLEC15 is the gene whereas Signlec-15 is the protein. If that's the case, please italicize the gene name so it's clear without any confusion.

2. Section 3.5 discusses the diagnostic significance of SIGLEC15 for molecular subtypes but has not been explained in detail. It would be beneficial to add some more context to this in the discussion to understand the significance of these findings.

3. Abbreviations should be kept consistent (for example, KM used for kaplan Meier curve used be used consistently in the document).

4. In section 3.11, were there any significant correlations found between Siglec-15 expression and the immune cell abundance/levels other than the co-expression? (Something similar to what has been done in section 3.12 with chi-square analysis.)

6. PLOS authors have the option to publish the peer review history of their article (what does this mean?). If published, this will include your full peer review and any attached files.

Reviewer #1: **Yes: **Parastoo Shahrouzi

Reviewer #2: No

---

## [Author Response · Author response to Decision Letter 0]

3 Sep 2024

We have revised our manuscript according to the reviewers comments and journal request.

---

## [Editor Report · Decision Letter 1]

9 Sep 2024

PONE-D-24-03046R1Expression of SIGLEC15 correlates with tumor immune infiltration, molecular subtypes, and breast cancer progressionPLOS ONE

Dear Dr. Chen,

Thank you for submitting your manuscript to PLOS ONE. After careful consideration, we feel that it has merit but does not fully meet PLOS ONE’s publication criteria as it currently stands. Therefore, we invite you to submit a revised version of the manuscript that addresses the points raised during the review process. Please address all the comments noted below.

We look forward to receiving your revised manuscript.

Kind regards,

Amy McCart Reed

Academic Editor

PLOS ONE

Additional Editor Comments:

Thank you for your resubmission.

I have reviewed in detail and find some additional areas that need addressing. Your manuscript would be greatly improved with English language editing. Please can you explain the need to add an additional author?

In the response to reviewers you say you didn’t use METABRIC, but it is in both the methods and the supplement?

Very unconventional statement to include in the introduction – please remove- Its role in cancer immune regulation was first characterized in 2019 by Prof. Chen [15].

How many cases did you do mIHC on? Was it also the TMA? Where is that data (to complement the images in Fig5) – do you have some co-expression plots?

Is there sufficient tumour associated normal in the TMA to support your conclusions about normal expression? A TMA typically focusses on a tumour rich region - an H&E would be helpful.

Table S1 – which datasets?

Table 2 – the first column is very confusing – it is not clear which variables are being analysed.

Figure S5 needs more annotation and higher powered view to support the subcellular localisation statements.

In the discussion you mentioned SIGLEC mRNA is prognostic in TCGA/METABRIC but you don’t discuss that is it not an independent prognostic marker as shown by your Cox multivariate analysis.

Remove the phrase ‘prognostic boon’ from the discussion.

This sentence – ‘Elevated SIGLEC15 expression is indicative of a potential Luminal subtype within the sample’ is not evidenced by your study. Please remove it.

---

## [Author Response · Author response to Decision Letter 1]

15 Oct 2024

Please check the respnse letter.

---

## [Editor Report · Decision Letter 2]

28 Oct 2024

Expression of SIGLEC15 correlates with tumor immune infiltration, molecular subtypes, and breast cancer progression

PONE-D-24-03046R2

Dear Dr. Chen,

We’re pleased to inform you that your manuscript has been judged scientifically suitable for publication and will be formally accepted for publication once it meets all outstanding technical requirements.

Kind regards,

Amy McCart Reed

Academic Editor

PLOS ONE

Additional Editor Comments (optional):

Thank you for your efforts to address the comments. There are still a couple of typos (eg. radiation in the supplement table) but you can correct these during proofing. You will also need to increase the size of the graphs in figure S3- the text is not legible.
---

## [Editor Report · Acceptance letter]

3 Nov 2024

PONE-D-24-03046R2 

PLOS ONE

Dear Dr. Chen, 

I'm pleased to inform you that your manuscript has been deemed suitable for publication in PLOS ONE. Congratulations! Your manuscript is now being handed over to our production team.

Kind regards, 

on behalf of

Associate Professor Amy McCart Reed 

Academic Editor

PLOS ONE